# BIOLOGICALLY INSPIRED SLEEP ALGORITHM FOR INCREASED GENERALIZATION AND ADVERSARIAL ROBUSTNESS IN DEEP NEURAL NETWORKS

**Timothy Tadros,**
Neurosciences Graduate Program
University of California, San Diego
La Jolla, CA
`tttadros@ucsd.edu`

**Giri P Krishnan,**
Department of Medicine
University of California, San Diego
La Jolla, CA

**Ramyaa Ramyaa,**
Department of Computer Science
New Mexico Tech,
Socorro, NM

**Maxim Bazhenov**
Department of Medicine
University of California, San Diego
La Jolla, CA
`mbazhenov@ucsd.edu`

## ABSTRACT

Current artificial neural networks (ANNs) can perform and excel at a variety of tasks ranging from image classification to spam detection through training on large datasets of labeled data. While the trained network may perform well on similar testing data, inputs that differ even slightly from the training data may trigger unpredictable behavior. Due to this limitation, it is possible to design inputs with very small perturbations that can result in misclassification. These adversarial attacks present a security risk to deployed ANNs and indicate a divergence between how ANNs and humans perform classification. Humans are robust at behaving in the presence of noise and are capable of correctly classifying objects that are noisy, blurred, or otherwise distorted. It has been hypothesized that sleep promotes generalization of knowledge and improves robustness against noise in animals and humans. In this work, we utilize a biologically inspired sleep phase in ANNs and demonstrate the benefit of sleep on defending against adversarial attacks as well as in increasing ANN classification robustness. We compare the sleep algorithm's performance on various robustness tasks with two previously proposed adversarial defenses - defensive distillation and fine-tuning. We report an increase in robustness after sleep phase to adversarial attacks as well as to general image distortions for three datasets: MNIST, CUB200, and a toy dataset. Overall, these results demonstrate the potential for biologically inspired solutions to solve existing problems in ANNs and guide the development of more robust, human-like ANNs.

## 1 INTRODUCTION

Although artificial neural networks (ANNs) have recently begun to rival human performance on various tasks, ranging from complex games (Silver et al. (2016)) to image classification (Krizhevsky et al. (2012)), ANNs have been shown to underperform when the testing data differs in specific ways even by a small amount from the training data (Geirhos et al. (2018)). This lack of generalization presents two issues when ANNs are utilized in the real world. First, ANNs are often trained on curated datasets of images designed to best capture the image content, whereas in real-world scenarios, they may be tested on disturbed or noisy inputs, not observed during training. Second, ANNs are susceptible to adversarial attacks, or the deliberate creation of inputs designed to fool ANNs that may be imperceptibly different from correctly classified inputs (Szegedy et al. (2013)). These two issues limit ANNs applicability in the real world and present potential security risks when deployed.

There have been two main approaches for investigating ANN robustness: adversarial machine learning and training data manipulation (Ford et al. (2019)). Adversarial machine learning aims to develop novel attack methods which perturb the input minimally while changing the ANN's classification outcome (Moosavi-Dezfooli et al. (2016); Carlini & Wagner (2017); Goodfellow et al. (2014); Athalye et al. (2017); Nguyen et al. (2015)) as well as to design defense mechanisms which prevent these attacks from affecting ANN behavior (Papernot et al. (2016b); Goodfellow et al. (2014); Huang et al. (2015), see Yuan et al. (2019) for review). Training data manipulation research typically examines the impact of changing the input distribution during testing and observing the effect on ANN performance. Geirhos et al. (2018) showed that ANNs trained on images with one type of distortion may not perform well when tested on other types of distortions, even if images with both distortions appear identical to the human eye. Likewise, ANNs trained on unperturbed images exhibit reduced performance when images in the test set are distorted, for example, through horizontal translations, blurring, or the addition of compression artifacts (Dodge & Karam (2016); Vasiljevic et al. (2016); Zhou et al. (2017)). Although it has been proposed that adversarial and manipulation robustness can be increased through various mechanisms during the training phase, such as fine-tuning, recent research has shown that these methods are mostly ineffective or their effectiveness is inconclusive (Geirhos et al. (2018); Uesato et al. (2018); Athalye et al. (2018)).

It has been hypothesized that in the mammalian brain sleep helps to create generalized representations of an input learned during the awake state (Stickgold & Walker (2013); Lewis & Durrant (2011)). Sleep has been identified as being critical for memory consolidation - a process of converting recent memories into long-tern storage (Rasch & Born (2013)). During sleep, there is reactivation of neurons involved in previously learned activity (Stickgold (2005)) and this reactivation is likely to invoke the same spatio-temporal pattern of neuronal firing as the pattern observed during training in the awake state (Wilson & McNaughton, 1994). Sleep reactivation, or replay, serves to strengthen synapses involved in a learned task through local synaptic plasticity, such as Spike Time Dependent Plasticity (STDP). Plastic changes during sleep can increase a subject's ability to form connections between memories and to generalize knowledge learned during the awake state (Payne et al. (2009)). In one study (Wamsley et al. (2010)), subjects learned to find an exit to a maze in a virtual 3D environment. Subjects who were allowed to sleep exhibited a more complex understanding of the overall shape of the maze (Wamsley et al. (2010)). Using biophysical model of a cortical network Gonzalez et al. (2019) and Wei et al. (2018) showed that sleep dynamics promotes reactivation and helps to create distinct representations for unique memories by devoting synapses to specific memory traces. This body of neuroscience work suggests that a sleep-like activity may be applied to ANNs to enable the network to extract the gist of the training data without being constrained by the statistics of a specific training data set. Our specific hypothesis is that sleep phase could aid in reducing a neural network's susceptibility to adversarial attacks and to increase generalization performance by reducing the impact that imperceptible input changes can have on the task output.

In this new work, we propose a sleep-inspired algorithm to defend against adversarial attacks as well as to increase ANN robustness to noise. We utilize the notion of sleep from biology and apply an off-line unsupervised "sleep" phase to modify the parameters of a fully connected ANN. We demonstrate a number of performance improvements over existing defense algorithms, such as fine-tuning or adversarial retraining and defensive distillation, on both adversarial and noise robustness. The contributions are summarized below:

- We analyze how robust the proposed sleep algorithm is to four different types of adversarial attacks on three different datasets (MNIST, CUB200, and a toy dataset). For most conditions (MNIST, toy dataset), after sleep phase was applied, the attacks consistently resulted in adversarial examples that were more distinct from the original input compared to the adversarial examples designed for the original (before sleep) network.
- We illustrate that the sleep algorithm creates a more robust network whereby performance on noisy and blurred inputs is higher compared to control or defensively distilled network and is more robust to the other types of distortions compared to ANNs that are fine-tuned on a single distortion.
- We analyze the impact of the sleep algorithm on task representation and demonstrate that the algorithm creates decision boundaries that more closely resemble the true classes, effectively extracting the gist of the data.

## 2   ADVERSARIAL ATTACKS AND DISTORTIONS

Adversarial attacks aim to create minimal perturbations that, while imperceptible to the human eye, fool ANNs. These attacks range from white-box to black-box attacks, based on how much information they assume the attacker to possess about the network. White-box attacks assume that the attacker has access to the network architecture, training data and weights. These attacks can range from absolute information, such as gradient-based attacks which compute the gradient of the loss with respect to the input (Brendel et al. (2017)), to score-based attacks which only utilize predicted scores of the model. Black-box attacks, which assume no knowledge about the network, solely rely on the decision made in order to craft adversarial examples. Attacks can be (a) targeted such that the attacker aims to create an adversarial example that the network would predict as a certain class or (b) untargeted where the attacker's goal is simply to cause any kind of misclassification (Biggio & Roli (2018)).

In this work we consider four types of adversarial attacks ranging from white-box to black-box attacks. We assume that the attacker solely wants to cause a misclassification, with no respect to the output class. We present a brief description of each of the four attacks below (see Appendix for examples of images created by these attacks).

**Fast Gradient Sign Method (FGSM).** FGSM (Goodfellow et al. (2014)) computes the sign of the gradient of the loss function ($J$) with respect to the original input $x$ using the weights $\theta$ of the network and the target labels $y$.

$$x' = x + \epsilon sign(\nabla_x J(\theta, x, y)).$$

This represents the direction to change each pixel in the original input in order to increase the loss function. Based on the value of $\epsilon$, the corresponding perturbation to the original image can range from small to large. Thus, in this work we use the average of the smallest values of $\epsilon$ needed to create an adversarial example $x'$ (misclassified input) for each input in the testing set.

**DeepFool.** DeepFool (Moosavi-Dezfooli et al. (2016)) is an iterative method which approximates the nearest decision boundary to the input at time $t$ and moves the input $x_t$ in that direction to compute $x_{t+1}$. This process is repeated until a misclassification is produced or the runtime of the simulation is exceeded. For this attack, we measure the L2-norm between the original input $x$ and the adversarial input $x'$. Thus, successful defenses should result in a high L2-norm for this algorithm.

**Jacobian-based Saliency Map (JSMA).** JSMA (Papernot et al. (2016a)) aims to craft adversarial examples that minimize the L0-norm of $x - x'$ by reducing the number of pixels that are altered. In summary, the algorithm computes the gradient, as done in FGSM but for all possible classes. These gradient values represent how changing each pixel contributes to the overall loss function, with large values indicating a significant effect on the loss. These values are used to create a saliency map, where each pixel's impact on the loss is modelled. The algorithm utilizes this saliency map to alter individual pixels, repeating the gradient and saliency map computation until an adversarial example is created. For this type of attack, we utilize the L2-norm to determine defense success.

**Boundary Attack.** The Boundary Attack (Brendel et al. (2017)) is a black-box attack which relies solely on the decision of the ANN to craft an adversarial example. Given an input $x$, a random input $x'_0$ is chosen such that $f(x) \neq f(x'_0)$, where $f(x)$ is the label produced by the ANN. In our work, $x'_0$ is chosen from a uniform distribution. The attack starts by moving $x'_0$ toward $x$ until it reaches the point where $f(x) = f(x'_0)$, or the decision boundary in between $f(x)$ and $f(x'_0)$. From here, the attack consists of two steps: an orthogonal perturbation and a forward perturbation. During the orthogonal perturbation, random points along the hypersphere around $f(x'_t)$ are sampled. Those that are adversarial and closer to $x$ than before are added to the queue for forward perturbation. During the forward perturbation, a small step is taken from $x'_t$ to $x$ as long as $f(x) \neq f(x'_t)$. This process is repeated until a convergence criterion is met. For this attack, we utilize the L2-norm to define defense success.

**Distortions.** Although not specifically designed to attack an ANN, distortions negatively impact ANN performance. In this work we consider two simple distortion techniques: blurring and Gaussian noise. In first case, we perform 2-D Gaussian filtering with a blur kernel of varying standard deviation in order to blur the images. In second case, we add Gaussian noise with mean 0 and standard deviation $\sigma$. These distortions are tested in the networks implementing the proposed sleep algorithm as well as using the adversarial defenses discussed below.

## 3 Adversarial Defenses

We compare our sleep algorithm with two existing adversarial defenses: defensive distillation and fine-tuning, or adversarial retraining. Defensive distillation (Papernot et al. (2016b)) utilizes two training sessions in order to create a distilled network. First, an initial network is trained on $(X, Y)$, where $X$ is the training data, and $Y$ is the one-hot encoded training labels. The activation function of this network is changed such that the softmax function of the output layer is computed using a temperature term $T$ as follows:

$$F(x) = \frac{e^{\frac{z_i(X)}{T}}}{\sum_{l=0}^{N-1} e^{\frac{z_l(X)}{T}}}.$$

A higher $T$ forces the ANN to produce larger probability values for each class, whereas lower $T$ values support a similar representation as the one-hot encoded labels. After the first network is trained, the output of the network (probability values) is used to train a distilled network with the same softmax-temperature function. Previous work has shown this approach can be successful at preventing some types of attacks (Papernot et al. (2016b)). However, others have shown that it is not successful at defending against modified versions of those attacks or novel attacks in general (Carlini & Wagner (2016; 2017)). Based on the previous work which found that temperature values between 20 and 100 effectively prevent adversarial attacks (Papernot et al. (2016b)), we use a temperature value of $T = 50$ in our implementation of defensive distillation.

Adversarial retraining aims to fine-tune the network on adversarial examples with the correct labels as a form of regularization. Previous work has shown that adversarial retraining can mitigate the effectiveness of some adversarial attacks. Goodfellow et al. (2014) showed that adversarial retraining can reduce the error rate on MNIST, demonstrating greater ANN robustness after fine-tuning. Likewise, Moosavi-Dezfooli et al. (2016) showed that fine-tuning on DeepFool attacks can reduce the effectiveness of their attacks. However, they observed that fine-tuning on FGSM attacks has negative results, actually increasing the strength of the attack. This suggests that fine-tuning may overfit the network to certain attacks, while failing to extrapolate to other attacks, similar to results shown for generalization in ANNs (Geirhos et al. (2018)). For the adversarial retraining procedure presented here, we train the network on the original input and then fine-tune the network on various adversarial attacks with a reduced learning rate.

## 4 Sleep algorithm

The basic intuition behind the sleep algorithm is that a period of offline activity, whereby network weights are modified according to an unsupervised learning algorithm, allows the parameters of the network to become more reflective of the underlying statistics of the task at hand, while not overfitting the statistics of the training data. The pseudocode is presented in Algorithm 1. In short, an ANN is trained using stochastic gradient descent and the standard backpropagation algorithm (exact parameters used for each of the datasets are shown in Table 2). After training, the network structure is converted into a spiking neural network (SNN). After building the SNN, we run a sleep phase which modifies the network connectivity based on spike-timing dependent plasticity (STDP). After the sleep phase, the SNN network is converted back into the ANN and testing is performed.

### 4.1 Spiking Neural Networks

SNNs seek to model closely temporal brain dynamics. In short, SNNs are composed of spiking neurons and model the information transformation and the dependence on exact timing of spikes that occurs in biological networks (Ghosh-Dastidar & Adeli (2009)). Individual neuron models can range from simple integrate-and-fire type neurons which sum their inputs and produce an output (spike) if this exceeds some firing threshold to more complex Hodgkin-Huxley type neurons which model sodium-, potassium-, and chloride-channel kinetics (Abbott & Kepler (1990)). Recent work has shown that a near loss-less conversion between ANNs and SNNs can be achieved by propagating activity through a spiking neural network for a given input and counting the number of times that each output neuron fires (Diehl et al. (2015)).

To convert an ANN to SNN (Lines 1-3 of pseudocode), we assume the ANN utilizes ReLU neurons with no bias. This assumption is made so that the output neuron's activation can be treated as a firing rate, either zero or positive, and that the thresholds of all neurons in a given layer are of the same scale. The weights from the ANN are directly mapped to the SNN. In our analysis, each unit in the SNN is modelled as an integrate-and-fire type neuron, computing the following equation:

$$\tau_m \frac{dv}{dt} = -v(t) + \sum_{i=1}^{N} w_i * s(i).$$

Here, $\tau_m$ represents the decay constant of the membrane potential, $v$ is the voltage at a given time, $w_i$ is the weight connecting from neuron $i$, and $s(i)$ is the spiking activity of neuron $i$, either 1 or 0.

## 4.2 Plasticity and Sleep

The key advantage of using a SNN is that biologically inspired training rules can be applied while the network is driven by noisy input. Empirical data suggest that the brain uses spike-timing dependent plasticity (STDP) (Song et al., 2000), where weight updates depend on the relative timing of pre- and post-synaptic spikes. It has been shown that STDP results in balanced activity, where all neurons fire in equal proportions (Song et al. (2000)). Here we utilize a modified version of STDP: if a pre-synaptic spike induces a post-synaptic spike, then the weight between these neurons is increased. If a post-synaptic spike occurs, but the pre-synaptic neuron does not spike, then the corresponding weight is decreased (in this case postsynaptic spiking may occur because of spiking in other neurons connecting to that post-synaptic neuron).

The sleep training phase we propose here can be described as following. First, inputs to each neuron of the input layer must be presented as spiking activity in order to propagate activity from the input layer to the hidden layers of the network. We convert inputs (real-valued pixel intensities or features) to spikes by defining a maximum firing rate $f_{max}$ with units $\frac{spikes}{sec}$ and computing a Poisson-distributed spike raster, such that inputs with higher values (i.e. brighter pixels) will have higher rate than inputs with lower values, with no spike rates exceeding $f_{max}$. Next, activity is propagated through the network as spikes and the STDP rule is applied to update weights. In biological networks, increase of synaptic strength during slow-wave sleep leads to characteristic patterns of activity with repetitive periods of elevated firing (Up-states), when previously learned memory traces are spontaneously replayed. To simulate this dynamics, synaptic weights in SNN are up-scaled to induce high firing rates in later layers. Other important parameters include the threshold for each layer and the length of sleep. The parameters used for each dataset are presented in Table 3.

## 4.3 Experiments and Datasets

Below, we describe the general experimental setup as well as the datasets tested. First, we trained a control ANN using the training set for each of the main datasets used in this study. Next, we created a defensively distilled network using $T = 50$ for the temperature parameter to create the second test network. Then, we fine-tuned the control ANN on a specific attack or distortion method to create the third test network. Finally, we converted the control ANN to an SNN and applied the sleep algorithm as described above to create the fourth test network. We created adversarial examples for each of these four networks using the attacks we described above (fine-tuned networks are tested on the attacks they were fine-tuned on). Then, we analyze how successful each attack is to fool each of the four networks using the metrics defined above. For generalization (blur and noise), we performed the same setup as above creating four different networks. We then tested each network on varying levels of distortion. We tested networks fine-tuned on blurred and noisy images to measure how performance generalizes across distortion methods. We averaged performance across a minimum of three networks for each attack and distortion.

We used three datasets to compare performance: Patches (a toy dataset created simply for analysis), MNIST (LeCun et al. (1998)), and CUB-200 (Welinder et al. (2010)). Patches consists of four binary images arranged in a 10x10 square. Each image has its own label (1-4), and consists of 25 bright pixels (value set to 1) and 75 dark pixels. The overlap of bright pixels among the four images (see Appendix] Appendix) is chosen such that the task is not trivial. The MNIST dataset consists of 70,000 28x28 greyscale images of handwritten digits, with 60,000 in the training set and 10,000 in the testing set. CUB-200 is a high resolution dataset of images of birds with 200 bird species,

---

**Algorithm 1 Sleep:**

1: **procedure** CONVERTANNTOSNN($nn$)
2:      Map the weights from ($nn$) with ReLU units to network of integrate-fire units ($snn$)
3:      Apply weight normalization and return scale for each layer ([24]) **return** $snn, scales$
4: **procedure** CONVERTSNNTOANN($nn$)
5:      Directly map the weights from integrate-fire network ($nn$) to ReLU network ($ann$) **return** $ann$
6: **procedure** SLEEP($nn, I, scales$)                          ▷ $I$ is input
7:      Initialize $v$ (voltage) $= 0$ vectors for all neurons
8:      **for** $t \leftarrow 1$ to $Ts$ **do**                   ▷ $Ts$ - duration of sleep
9:         $S(1, t) \leftarrow$ Convert input $I$ to Poisson-distributed spiking activity
10:         **for** $l \leftarrow 2$ to $n$ **do**             ▷ n - number of layers
11:            $v(l, t) \leftarrow v(l, t-1) + (scales(l-1)\mathbf{W}(l, l-1)S(l-1, t))$    ▷ W(l,l-1) - weights
12:            $S(l, t) \leftarrow v(l, t) > threshold(l)$              ▷ Propagate spikes
13:            $\mathbf{W}(l, l-1) \leftarrow \begin{cases} \mathbf{W}(l, l-1) + inc & \text{if } S(l,t) = 1 \,\&\, S(l-1, t) = 1 \\ \mathbf{W}(l, l-1) - dec & \text{if } S(l,t) = 1 \,\&\, S(l-1, t) = 0 \end{cases}$    ▷ STDP

14: **procedure** MAIN
15:      Initialize neural network ($ann$) with ReLU neurons and bias $= 0$.
16:      Train $ann$ using backpropagation.
17:      $snn, scales = $ ConvertANNtoSNN($ann$)
18:      $snn = $ Sleep($snn$, Training data $X$, $scales$)
19:      $ann = $ ConvertSNNtoANN($snn$)

---

with very few ( 30) images per class. For this dataset, we used previously extracted ResNet-50 embeddings, where ResNet-50 was pre-trained on ImageNet (He et al. (2016)). For CUB-200, we do not report results for blurring, since we are using extracted features, not images.

## 5 RESULTS

We evaluate the sleep algorithm in two settings: (1) Adversarial attacks designed to fool neural networks and (2) generalization distortions designed to reflect imperfect viewing conditions or other types of noise. For adversarial attacks (other than FGSM), we utilize the following metric to evaluate the success of each defense. Let $x_i'$ be the adversarial example created for input $x_i$. The total score $S_A$ for an attack is the median squared L2-distance for all samples, where $N$ is the dimension of the space:

$$S_A = median(\frac{1}{N} \|x_i' - x_i\|_2^2).$$

For FGSM, we define the following metric which computes the median minimum noise level $\epsilon$ needed to produce a misclassification across all samples:

$$S_{FGSM} = median(min(\epsilon_i)) \ s.t. \ f(x_i + \epsilon_i * x_i') \neq f(x).$$

For MNIST and CUB-200, we evaluate the attacks on all examples in the testing set. Examples that the networks get wrong before the attack was implemented are discarded from the analysis (in these cases $\|x_i' - x\|_k^2 = 0$ and $\epsilon_i = 0$ for all attacks). For FGSM and distortions, we also include plots of classification accuracy as a function of noise level. For DeepFool and JSMA, we report adversarial attack accuracy (number of examples where $f(x) = y$ and $f(x') \neq f(x)$, where $y$ is the correct label, over number of examples tested). Note that these algorithms would always produce an adversarial example if allowed to run forever. However, due to computational limitations, we included a run-time limits on the number of iterations for these algorithms (see Appendix). Thus, a lower adversarial attack accuracy indicates that the attack would need more iterations to run to reach 100% accuracy. This is a similar measure as distance since more iterations would result in more distinct adversaries for all attacks implemented and the updates at each iteration have the same magnitude for each defense.

| Patches | Control | Defensive Distillation | Fine-tuning | Sleep |
|---|---|---|---|---|
| FGSM | 0.0175 | 0.05 | 0.1425 | **0.2025** |
| DeepFool | **0.0440**(95.00%) | 0.0360 **(90.00%)** | 0.0201 (100.0%) | 0.0124 (100.0%) |
| JSMA | 0.0049 (80.00%) | 0.0135 **(70.00%)** | 0.0450 (100.0%) | **0.0541** (100.0%) |
| Boundary Attack | 0.2971 | 0.3124 | 0.1772 | **0.3515** |

| MNIST | Control | Defensive Distillation | Fine-tuning | Sleep |
|---|---|---|---|---|
| FGSM | 0.0900 | 0.0900 | 0.1000 | **0.2200** |
| DeepFool | 0.0042 (96.46%) | 0.0043 (96.42%) | 0.0074 (97.40%) | **0.0484 (86.38%)** |
| JSMA | 0.0090 (99.29%) | 0.0086 (99.41%) | **0.0133** (98.77%) | 0.0059 **(72.97%)** |
| Boundary Attack | 0.0525 | 0.0525 | **0.0544** | 0.0488 |

| CUB-200 | Control | Defensive Distillation | Fine-tuning | Sleep |
|---|---|---|---|---|
| FGSM | 0.0550 | 0.0500 | **0.0650** | 0.0600 |
| DeepFool | 0.0027 **(82.23%)** | 0.0019 (84.16%) | **0.0044 (83.02%)** | 0.0025 (83.12%) |
| JSMA | 0.0477 (95.56%) | 0.0347 (95.88%) | **0.0530** (95.38%) | 0.0439 **(95.15%)** |
| Boundary Attack | 0.9751 | 0.9034 | **0.9976** | 0.9967 |

Table 1: Adversarial Attack Scores (Best defense scores are bolded, lowest attack success rates are in blue)

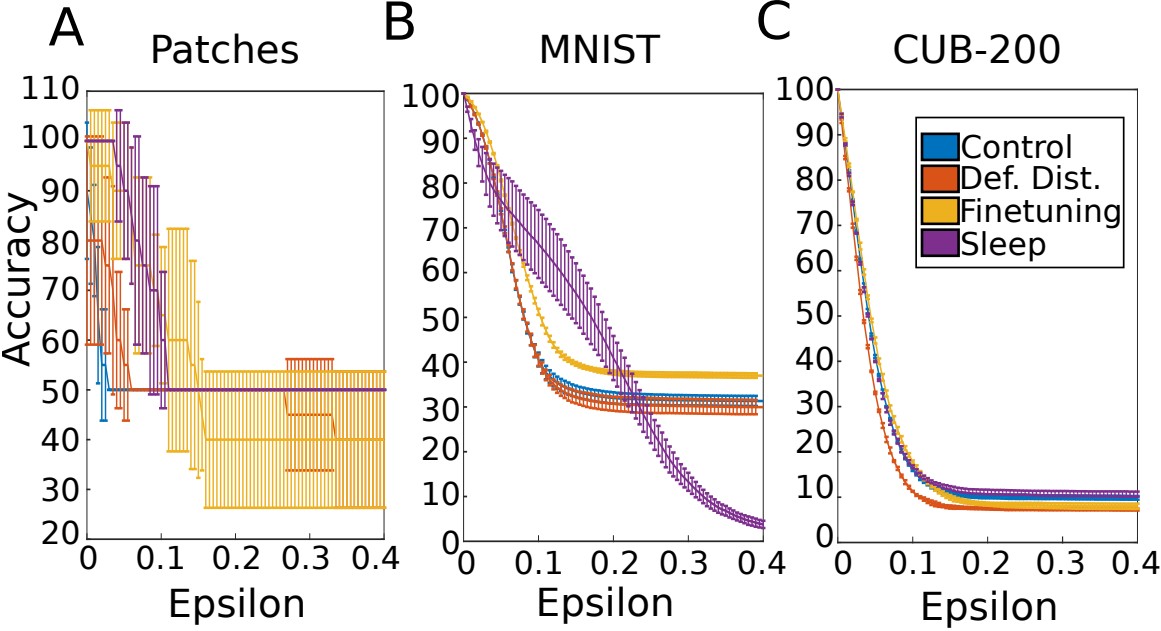

Figure 1: FGSM classification accuracy as a function of noise added (Epsilon) for the three datasets tested: A) Patches B) MNIST C) CUB-200.

## 5.1 ADVERSARIAL ATTACKS

Here we report the scores for all different attacks and for the all datasets. For the FGSM attack, the sleep algorithm increases the median minimum noise needed for misclassification for all three datasets compared to the control network (also see Figure 1). For MNIST dataset, the amount of noise needed to fool the network after the sleep algorithm was almost double of that needed for either the fine-tuning or defensive distillation approaches. For the Patches dataset, both defensive distillation and fine-tuning increase the robustness of the network. However, on CUB-200, only fine-tuning and sleep were able to defend, albeit marginally, against the FGSM attack. Looking at the classification accuracy of the network as a function of noise added ($\epsilon$, Figure 1), we observe that

in the Patches and CUB-200 dataset, sleep tends to have higher classification accuracy than the other methods for epsilon $< 0.1$. After this point, sleep tends to have equal classification accuracies as compared to the other methods. For MNIST, the baseline classification accuracy on the original test set decreases slightly compared to the other methods (80% after sleep). However, the performance remains high longer than for the other defense methods on images that were correctly classified. We observed that performance continued to drop after a sufficiently large amount of noise was added. This is biologically plausible as adding more noise to an image should result in image degradation and misclassifications. In sum, these results indicate that a sleep phase can successfully mitigate FGSM, more so than a control network.

For DeepFool, sleep has a significant effect on the defense score on the MNIST dataset, both reducing the attack success rate and increasing the distance between the adversarial example and the original input by an order of magnitude. For Patches and CUB-200 this effect is less pronounced, with fine-tuning or the control network performing better. We hypothesize that sleep was ineffective in preventing the DeepFool attack in tasks with very few exemplars per class (Patches) or a large number of classes (CUB-200). In CUB-200, there is a large number of classes so the distance between the input and the nearest decision boundary is smaller (this is supported by the fact that JSMA, an L0 attack, does worse than DeepFool for CUB-200 and vice versa for MNIST, control networks). In this case, sleep is unable to move the decision boundary of one class without impinging on the decision space of another class. In MNIST, where the decision space for one class is presumably larger, sleep can alter decision boundaries in a way that has a minimal effect on other classes.

Sleep successfully increases the network's robustness to the JSMA attacks on MNIST and Patches, reducing the attack success rate in the case of MNIST and increasing the distance needed to create an adversary for Patches. On CUB-200, there is a marginal reduction in the adversarial attack accuracy compared to the control network. Defensive distillation and fine-tuning also reduce JSMA's effectiveness. However, for these two defenses, in the case of MNIST, the networks were capable of finding an adversary for a higher percentage of the testing set. Thus, the effect of changing a small number of important pixels is mitigated after running the sleep algorithm.

For the Boundary Attack, we found that no defense mechanism helps compared to the control in decreasing the attack's effectiveness on the MNIST dataset. However, for CUB-200 and Patches, the sleep algorithm results in a higher defense score than that for the control network. This lends support to the idea that sleep favorably alters decision boundaries so that it becomes harder to find an adversarial example that is close to the original image after the sleep phase. This also suggests that sleep is not simply obfuscating gradients, which has been a common criticism of several adversarial defenses (Athalye et al. (2018)), which are tested on white-box attacks. In fact, given the long run-time for convergence of this algorithm, if we define a threshold for adversarial attack success (L2-norm $> 1$), then sleep successfully defends against this attack on the MNIST dataset (see Table 4).

Why does sleep phase help? It has been shown that sleep tends to promote an increase in stronger weights while pruning weaker weights, thus increasing the width of the weights' distribution (Gonzalez et al., 2019). This results in the consolidation of strong memories at the cost of diminishing weak memories. From this point of view, a memory is a subspace or abstraction in the decision space corresponding to a given class. Sleep may result in enlarging the subspace the network allocates to a stronger category while shrinking weaker ones (Figure 5A). The process of strengthening the strong memory also results in making it robust and noise invariant, as seen in Figure 5B where the first 8 categories (numbers 0-7) are strengthened and become more invariant to the FGSM attack, while the last two digits are essentially forgotten and the network cannot confidently predict exemplars from these classes (Figure 5C). If the noise is less targeted, as in the case of random noise or blurring, sleep does not need to alter the decision space as much to produce better generalization and can maintain a high baseline accuracy, as we demonstrate in the next section.

## 5.2 GENERALIZATION

Figure 2 shows the network performance for noisy and blurry distortions of data for MNIST (A) as well as noisy distortions for the CUB-200 feature embeddings (B, see Figure 3 for results on Patches). Overall, fine-tuning on an image distortion results in the best performance for that specific distortion. However, as was noted (Geirhos et al. (2018)), fine-tuning on a specific distortion does

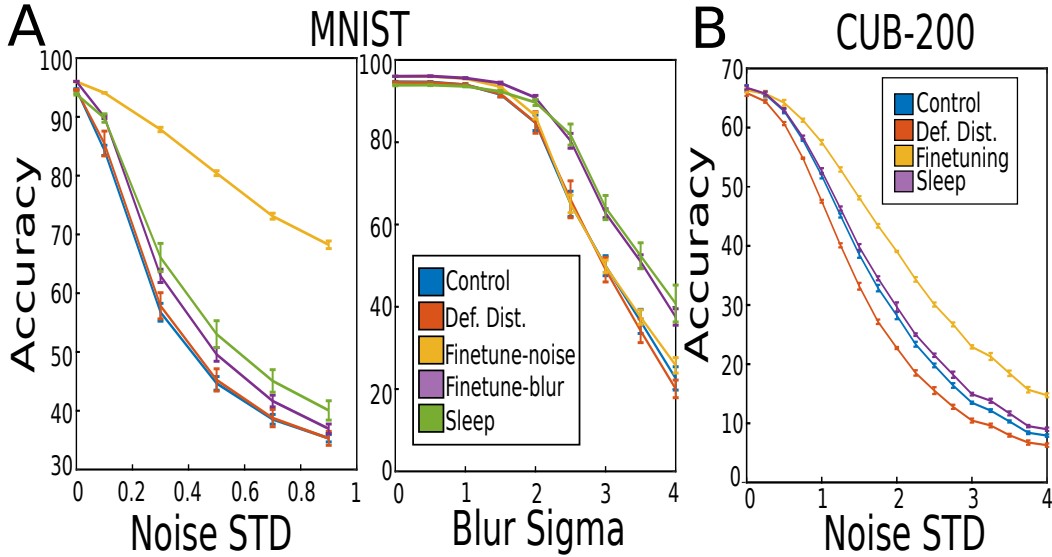

Figure 2: Sleep increases robustness to general distortions. A) Generalization classification accuracy for 5 networks for noise and blur on the MNIST dataset. B) Generalization classification accuracy for 4 networks for the noisy CUB-200 task. Note that there are only 4 networks because there is no blur task here.

not extend to other types of distortions. In our analysis, fine-tuning the network on blurred MNIST images results in high performance ($> 80\%$) on blurred images. However, for noisy images, this performance was only marginally above the control network. The sleep algorithm increased performance for both distortion methods, since this approach is not tailored to any one representation of the training set.

Finally we tested how sleep increases robustness on blur and noise distortions. In biological systems, sleep increases generalization through replay of memories learned during awake which leads to changes in synaptic weights. These changes entail both an increase in synaptic weights associated with a specific task and pruning of synapses involved in other tasks (Gonzalez et al., 2019; Tononi & Cirelli, 2006). Figures 9 and 10 show that correlations among like digits in the hidden layers of the network are greater after applying sleep than before for noisy and blurred images. Likewise, pairs of different digits usually become decorrelated after sleep, suggesting synaptic pruning. We also show that both normalized spiking activity and activations of digit-specific neurons are higher after sleep than before (Figures 11 and 12, see Appendix for details). These results suggest that the sleep algorithm increases robustness through biologically plausible learning mechanisms involving replay of relevant activity during sleep phase.

## 6 Conclusions and Future Directions

In this work, we show that a biologically inspired sleep algorithm can increase an ANN's robustness to both adversarial attacks and general image distortions. The algorithm augments the normal (e.g., back-propagation based) training phase of an ANN with an unsupervised learning phase in the equivalent SNN modelled after how the biological brain utilises sleep to improve learning. We hypothesize that the unsupervised sleep phase creates more natural feature representations which in turn lead to more natural decision boundaries, thus increasing the robustness of the network. Although this robustness may come at a cost of overall accuracy, it has been shown that robustness may have multiple important benefits, such as more salient feature representations as well as invariance to input modifications (Tsipras et al. (2018)). We also show that the trade-off between robustness and accuracy does not always occur, particularly for image distortions such as noise or blur. Future work includes converting the sleep algorithm into a regularization technique to be applied in more

standardized machine learning frameworks as well as understanding the theoretical basis for the beneficial role of spike based plasticity rules in increasing network robustness.

## 7    ACKNOWLEDGEMENTS

This work was supported by the Lifelong Learning Machines program from DARPA/MTO (HR0011-18-2-0021) and ONR (MURI: N00014-16-1-2829).

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

# 8 APPENDIX

## 8.1 TRAINING PARAMETERS

Here, we define the neural network parameters used for each of the three datasets as well as the sleep, defensive distillation, and fine-tuning parameters. Table 2 shows parameters used to train each of the control networks discussed in the paper. All neural networks were trained with ReLU neurons. Table 3 shows the parameters used during sleep for each of the three datasets. Note that these parameters for MNIST and CUB-200 were chosen by running a genetic algorithm to maximimize performance on the FGSM attack (performance was determined based on the training set so as not to overfit to the test set). For the other three attacks, parameters that maximized FGSM performance were used. Also, for noise and blur generalization, different parameters were chosen (not shown here).

|  | Patches | MNIST | CUB200 |
|---|---|---|---|
| Architecture | [100, 4] | [784, 1200, 1200, 10] | [2048, 350, 300, 200] |
| Learning Rate | 0.1 | 0.1 | 0.1 |
| Momentum | 0.5 | 0.5 | 0.5 |
| Dropout | 0 | 0.2 | 0.25 |
| Epochs | 1 | 2 | 100 |

Table 2: Parameters used to train the control network for each of the three datasets. Architecture refers to number of units per layer. For example, the MNIST network possessed 1 input layer, 2 hidden layers with 1200 units, and an output unit with 10 units.

|  | Patches | MNIST | CUB200 |
|---|---|---|---|
| Input Rate | 16 Hz | 40 Hz | 79 Hz |
| Sleep Duration | 3000 | 27105 | 11751 |
| Thresholds | 1.0450, 0.7150, 0.3850 | 36.18, 23.36, 36.38 | 2.69, 4.61, 2.63 |
| Synaptic AMPA current | 4.25 | 2.19 | 4.15 |
| Increase factor | 0.0035 | 0.063 | 0.0016 |
| Decrease factor | 0.0002 | 0.069 | 0.000209 |

Table 3: Parameters used during sleep. Input rate = $F_{max}$, the maximum firing rate of input neurons, Sleep duration = length of sleep (number of images presented during sleep, Thresholds = neuronal firing thresholds for each layer of neurons, Synaptic AMPA current = factor to scale the weights by during sleep, Increase and Decrease factor = amount weights are modified on a STDP event.

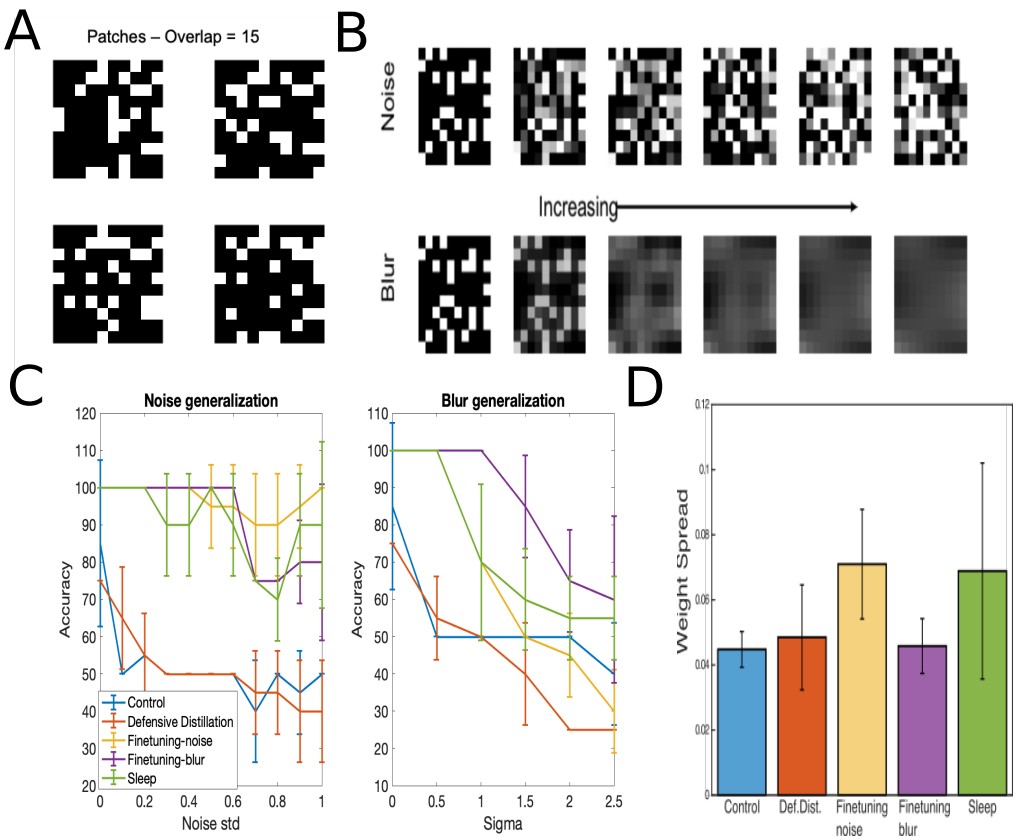

Figure 3: A) Patches dataset example - four binary images with 25 pixels turned on in each image and 15 pixel overlap. B)Types of images tested on for generalization for the Patches dataset. Top - Images with Gaussian noise added with increasing variannce (from 0 to 1.0 in steps of 0.2). Bottom - Gaussian blurred images with increasing sigma (from 0 to 2.5 in steps of 0.5). C) Generalization accuracy for noise and blur of five different networks tested (Control, Defensively distilled, Fine-tuned-blur, Fine-tuned-noise, and Sleep) for the Patches dataset. D) Weight spread for each of the 5 networks tested in C.

For the defensively distilled networks tested in the paper, we first train an initial network using a temperature of 50. Then, we use the training set to compute soft labels and finetune the initial network on these soft labels for the same number of epochs and with the same learning rate.

For the fine-tuned networks, we take the control networks trained with the parameters shown in Table 2. The learning rate is reduced to 0.05 and the network is fine-tuned on a mixture of either adversarial attacks, blur or noise and the original images/features. For CUB-200, we perform fine-tuning for 10 epochs.

## 8.2 PATCHES ANALYSIS

The Patches dataset represents an easily interpretable example where we can understand what happens to the weights after sleep. Figure 3A shows an example of the dataset. Here, we have 4 images each belonging to 4 different classes. 25 pixels are whitened in each image and the remaining 75 pixels are dark. There is a 15 pixel overlap, so that weights connecting from input to output layer must take this into account in order to separate the images. Figure 3B illustrate the blur and noise distortions tested for this dataset and Figure 3C shows the results for the blur and noise distortions.

After the network is trained, we can analyze the weights connecting from each of the 100 input neurons to the 4 output neurons (see Figure 4, top row). We theorize that optimally robust behavior would occur when weights connecting from ON-pixels are positive, weights connecting from

overlapping pixels are near 0, and weights connecting from OFF-pixels are negative. In this case, changing the value of overlapping pixels will have no effect on classification. Changing the value of OFF-pixels will cause the network to predict another class, where OFF-pixels may be ON-pixels or indicative of that class. Changing the value of ON-pixels will only have a negative impact if the brightness of the pixel is reduced significantly. Thus, in this circumstance, the network should behave robustly.

In the control network, we observe that weights connecting from ON-pixels (pixel-value = 1) increase while weights connecting from OFF-pixels remain at 0. Weights connecting from overlapping pixels remain near 0 or positive. Defensive distillation causes some weights connecting from overlapping pixels to decrease, likely because the soft labels used in defensive distillation cause overlapping pixel units to alter the probability values computed by the network in such a way that does not truly reflect the impact of the overlapping pixels. In the fine-tuned networks (both on blurred images and noisy images), we observe an increase in ON-pixel weights and an increase in noisiness of OFF-pixel weights. Likewise, in the sleep network, OFF-pixel weights become negative while ON-pixel weights remain the same. In these cases, robustness is increased as weights become more similar to our hypothesized ideal weights. Essentially, the magnitude of input changes need to change classification increase since the spread between ON-pixel weights and OFF-pixel weights increases. We quantify the spread in weights by taking the difference between the average weight connecting from ON-pixels and the average weight connecting from OFF-pixels. This represents the mean input that each correct output neuron receives. This result is shown in Figure 3D. Of note is that this weight spread is increased for both the sleep and finetuning-noise network, suggesting that these defenses bring the weights closer to their ideal values for computing robustness.

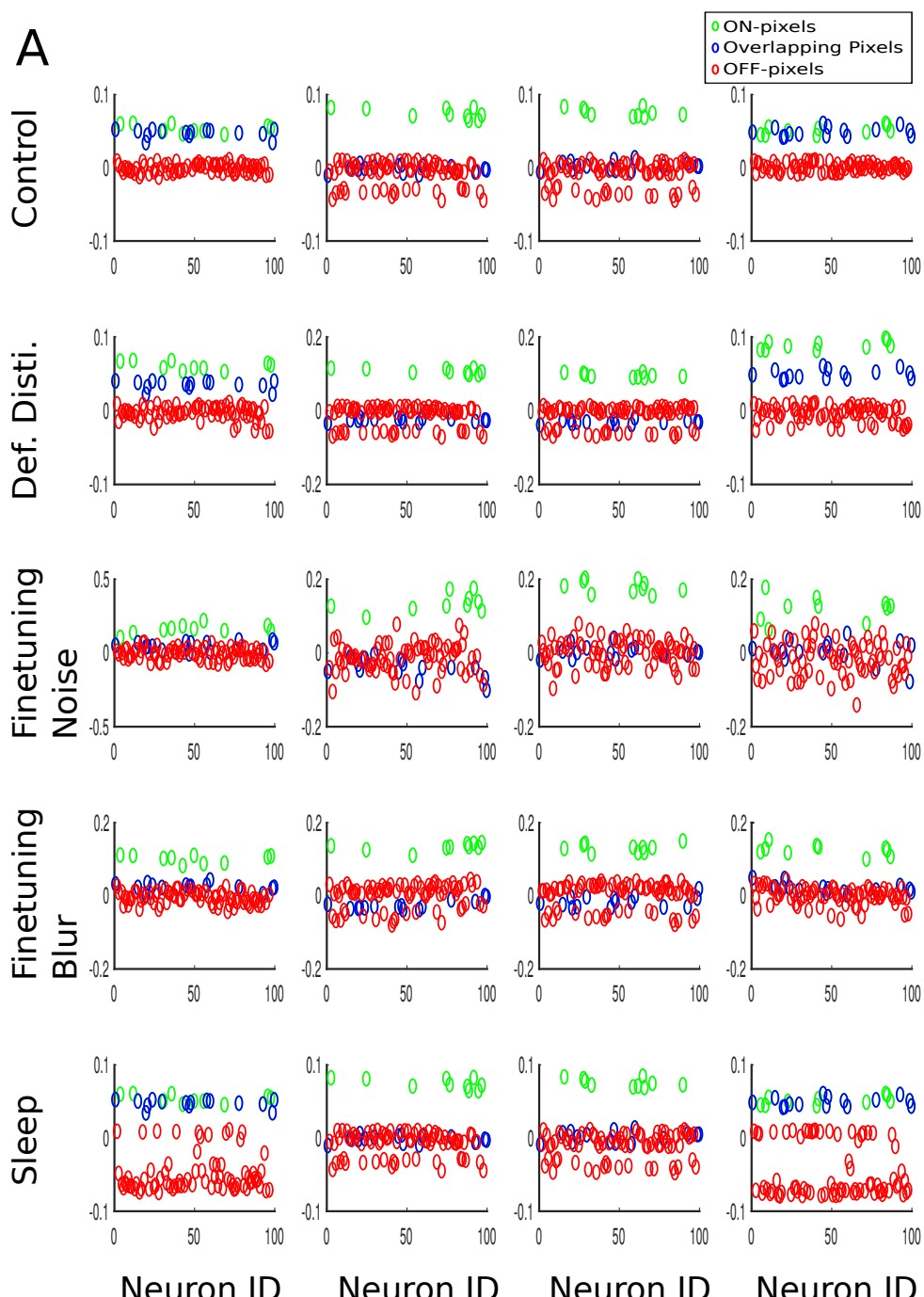

Figure 4: Weights for each of the 5 networks (Control, defensively distilled, finetuning on noise, finetuning on blur, and sleep) trained on the Patches dataset. Each column shows the weights value (y-axis) connecting from each of the 100 input neurons (x-axis) to the corresponding output neuron (i.e. the first column of graphs if the weights connecting from all input neurons to the first output neuron. Points are color-coded based on which of these pixels in the input layer correspond to ON-pixels, OFF-pixels, or overlapping pixels based on the input that should trigger that output neuron.

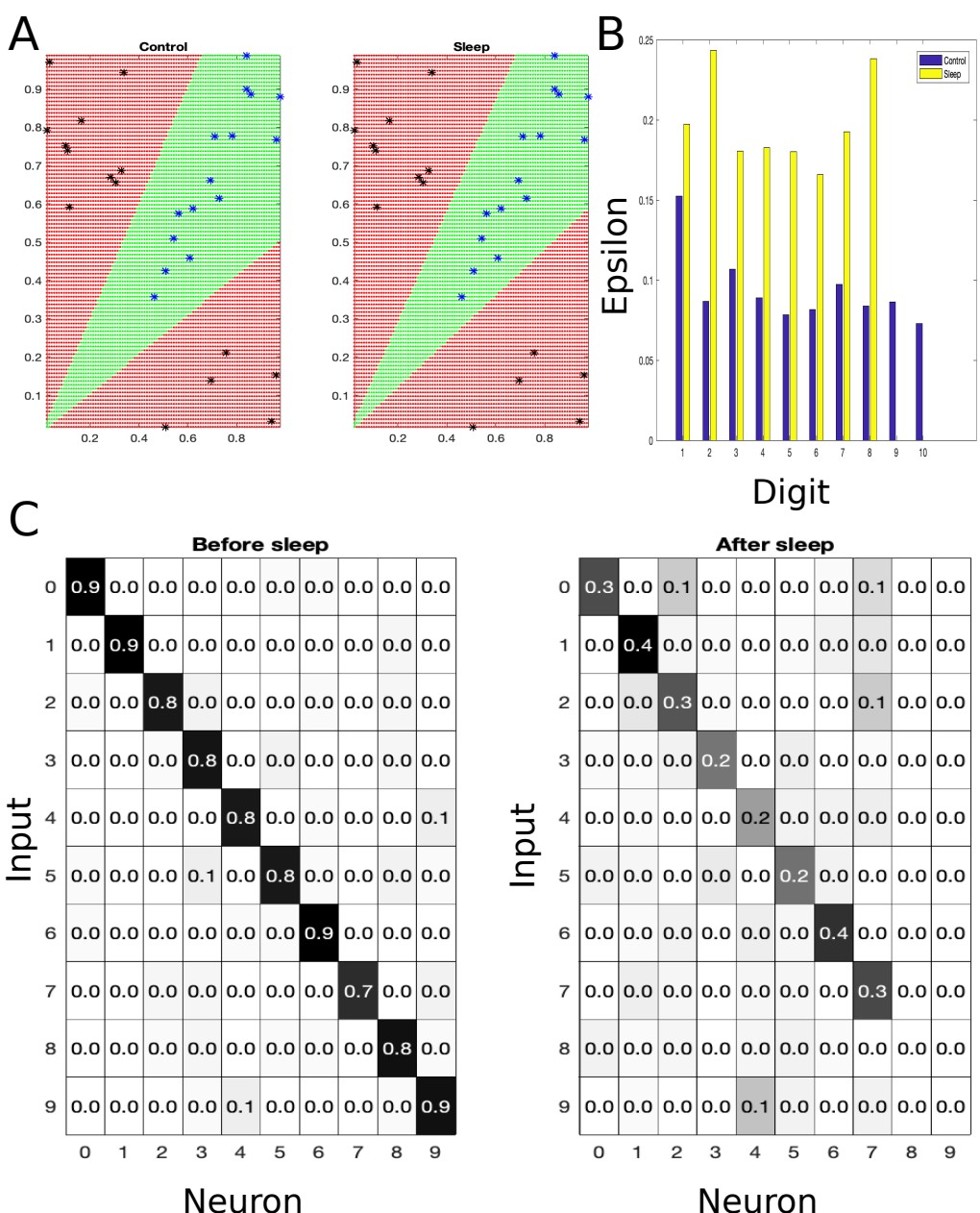

Figure 5: A) Example function learned in a 3-layer neural network illustrates that sleep alters decision boundaries in favor of making one class (corresponding to black points) more robust while impinging on another class (blue points). B) Average noise (epsilon) needed for FGSM attack for specific digits. C) Output layer scores for each digit (rows) before and after sleep. Columns represent average activation of each of the 10 output neurons.

## 8.3 ADVERSARIAL ATTACKS

Here, we describe the general approach for implementing DeepFool, JSMA, and the Boundary Attack discussed in the paper. We also show examples of adversaries created for each of the defense networks from these attacks.

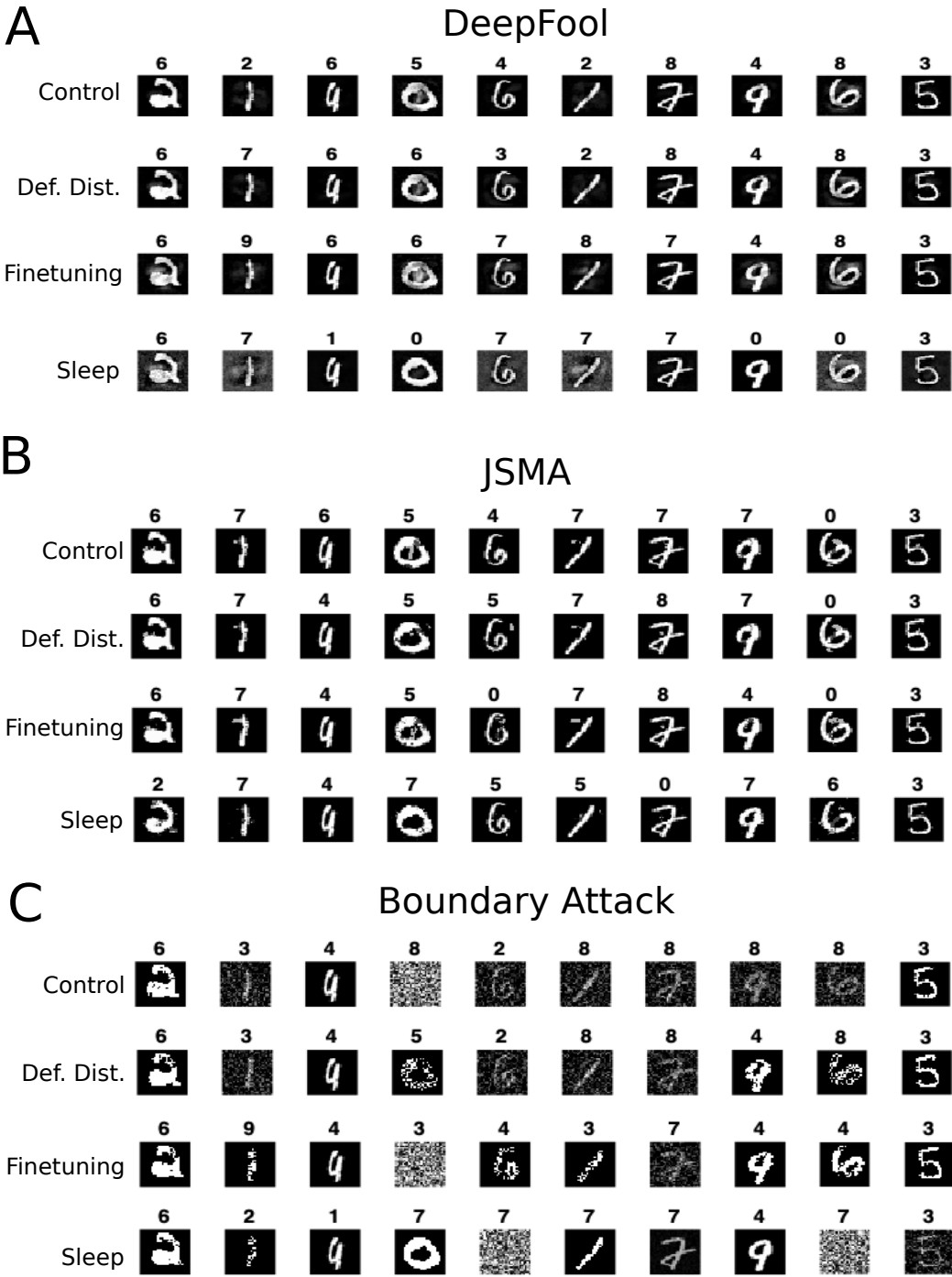

Figure 6: A) DeepFool adversarial examples for each defense. The network's prediction is shown above each image. B) JSMA. C) Boundary Attack.

**DeepFool.** DeepFool (Moosavi-Dezfooli et al. (2016)), as mentioned above, is an iterative algorithm that, at each iteration, aims to move the adversarial example in the direction of the closest decision boundary until it results in a misclassification. We based our implementation of that in Rauber et al. (2017). We stopped running the algorithm when either an adversarial example is found or when 100 iterations have passed. Examples of DeepFool attacks on the MNIST dataset are shown in Figure 6A. At each iteration we compute a linear approximation of the loss function and take a step in the

direction that would be result in a misclassification. The equations used and pseudocode can be found in the original DeepFool paper.

**JSMA.** JSMA is also an iterative algorithm which computes the pixel that would change the loss function the most at each algorithm and changes this pixel, until a misclassification is produced. For this method, we set a run-time limit of 500 iterations. We also remove a pixel from the saliency map when it has beeen updated seven times, so the algorithm can focus on other pixels. We set the change to each pixel at a constant value, 0.1. This represents how much each pixel is updated (in the direction that results in a misclassification) at each iteration. Pseudocode can be found in the original publication (Papernot et al. (2016a)). We show examples of adversaries created by JSMA in Figure 6B.

**Boundary Attack.** The Boundary Attack (Brendel et al. (2017)) starts with an adversarial example and moves it closer to the decision boundary of the correct class. At each step of the algorithm, the method performs orthogonal and forward perturbations to move the adversary closer to the original image, thus reducing the distance between the adversary and the original image. We set both a distance convergence criterion (L2-norm = 1e-7) and a run-time limitation on the attack (1000 iterations). Example attacks are shown in Figure 6C. We note that sometimes the algorithm does not successfully produce an "imperceptible" adversarial example and instead produces a noisy output (the starting condition is a noisy image). If we define a threshold defining a successful adversarial attack (L2-Norm > 1), then we observe the results for MNIST in Table 4.

| MNIST | Control | Defensive Distillation | Fine-tuning | Sleep |
|---|---|---|---|---|
| Boundary Attack | 0.0073 | 0.0074 | 0.0047 | **0.0094** |

Table 4: MNIST Boundary Attack with a threshold defining a successful adversarial attack.

## 8.4 GENERALIZATION ANALYSIS

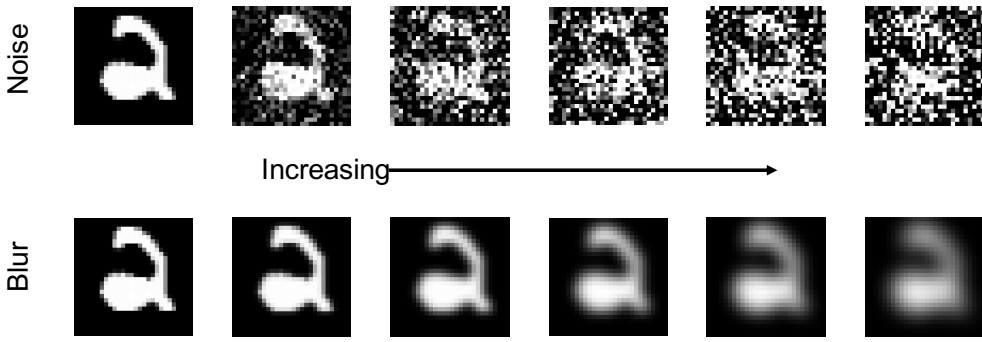

Figure 7: Types of images tested on for generalization for the MNIST dataset. Top - Images with Gaussian noise added with increasing variannce (0, 0.1, 0.3, 0.5, 0.7, 0.9, left to right). Bottom - Gaussian blurred images with increasing sigma (from 0 to 2.5 in steps of 0.5).

In this section, we analyze how sleep can aid in increasing ANN robustness. In biological networks, sleep extracts the gist of a task through replay (Lewis & Durrant (2011)). We hypothesized that our sleep algorithm works in the same manner. First, we tested the ability of the sleep network to decorrelate distinct inputs by analyzing the effect of running sleep and testing on our two distortion techniques (see Figure 7).

We computed the correlations of network activities in each of the hidden layers of the network before and after implementing our defense methods. For each pair of digits, we computed the average correlation of layer activities in the undistorted (Figure 8), noisy (Figure 9), and blurred (Figure 10) conditions. Each figure reports the difference in digit pairwise correlations between

the defense method and the control network for each set of inputs. For our sleep network, it is apparent that in layer 2 and layer 3, the correlations of the same digits (the diagonal) increases after sleep. Additionally, the correlation of distinct digits typically experiences negative change, representing decorrelation of distinct inputs. This analysis holds for defensive distillation and both of the fine-tuned networks. This suggests that the ANN representation of different exemplars of the same digit becomes more similar after sleep or after any of the defense networks when compared to the control. This is not simply due to an increased overlap of all inputs, since exemplars of different digits become decorrelated after applying a defense method.

Next, we performed the same correlation analysis on noisy and blurred images to see how the representation of distorted images changes after applying a distortion method. First, we note that fine-tuning on noisy images results in stronger correlation of the same (noisy) digit but weaker correlations of different (noisy) digits, as noted above. However, fine-tuning on blurred images does not have as strong an effect. Second, sleep seems to have a beneficial effect on the correlation matrices for both blurred and noisy images (comparing the right column of Figures 9 and 10). This illustrates the beneficial role of sleep in creating distinct representations of digits, where different neuronal ensembles encode different digits. This change in representation should result in increased robustness since changes to the input must be larger in order to recruit neuronal ensembles that represent other digits.

On top of decorrelating the representation of distinct memories by pruning synapses, biophysical modelling suggests that sleep can also aid in strengthening connections thus making stronger the response of primary neurons involved in memory recall (Gonzalez et al. (2019)). To test this hypothesis in our networks, we analyzed the firing rate and activations of digit-specific neurons before and after sleep. Before describing the analysis, we would like to note that SNNs can be used to performn classification and a near loss-less conversion between ANNs and SNNs has been achieved on the MNIST task (Diehl et al. (2015)). To perform classification, a digit is presented (as a Poisson spike train) to the network and spikes are propagated throughout the network for a given time period (or number of presentations of the input). Analyzing network activity in the spiking domain can be easier than in the activation domain (ANNs) since spikes are oftentimes easier to interpret than neuronal activations.

For this reason, we first analyze how spike rates of digit-specific neurons change before and after sleep in the spike domain. To do this we present all images of a specified digit to the spiking network and count the number of spikes from each neuron (holding the weights constant). We define digit-specificity by looking at the 100 neurons with the highest firing rates in layer 2. In Figure 11, we show that the normalized firing rate of these neurons usually increases after sleep (normalized by dividing by the maximum firing rate observed from the SNN).

Next, we perform the same analysis in the activation domain. Again, we define digit-specific neurons by looking at the top 100 neurons with the highest activation for a specific digit. We look at the normalized mean activations of these neurons before and after sleep and note that for all digits this value is higher after sleep than before sleep (Figure 12). This suggests that the neurons in the network are responding more strongly to the presentation of the same digit, thus increasing the robustness of the network as more noise must be added in order to counter the effect of this stronger response. This also suggests that our algorithm works in a biologically plausible way: both by decorrelating distinct inputs and increasing the strength of similar inputs.

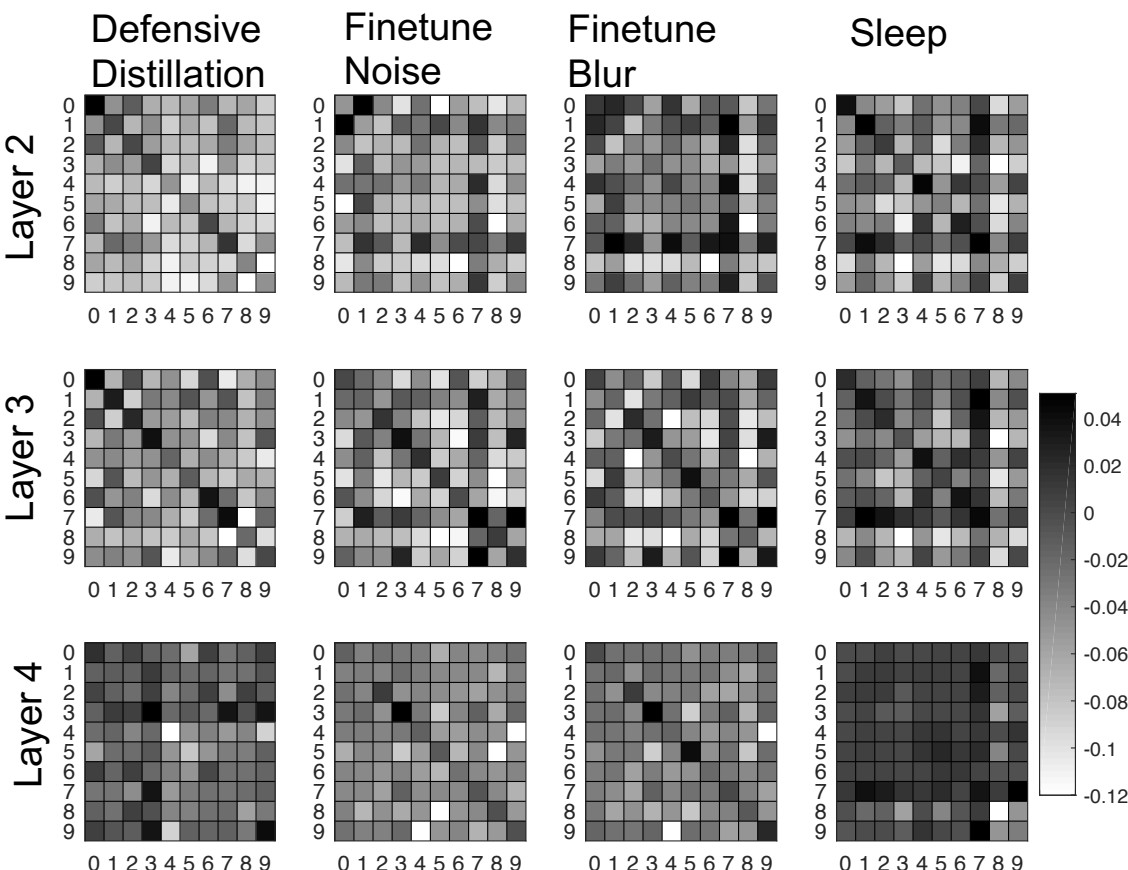

Figure 8: Correlation differences between defense network and control network for 4 different defenses. Correlations are computed based on the activations in each layer for each pair of digits (mean correlation). The difference between the correlation of the defense method (column) and the control network is plotted. Activations are computed based on the undistorted test images.

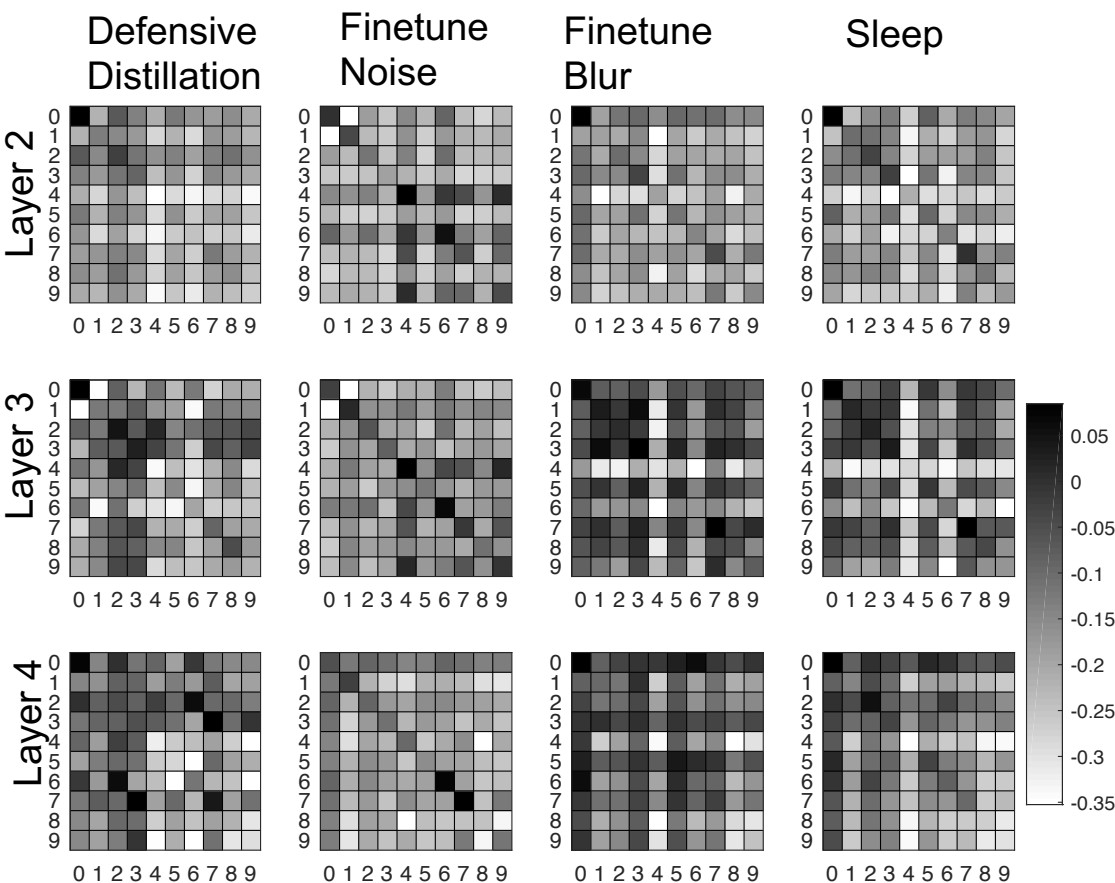

Figure 9: same as Figure 8 but activations are computed when presented with noisy images.

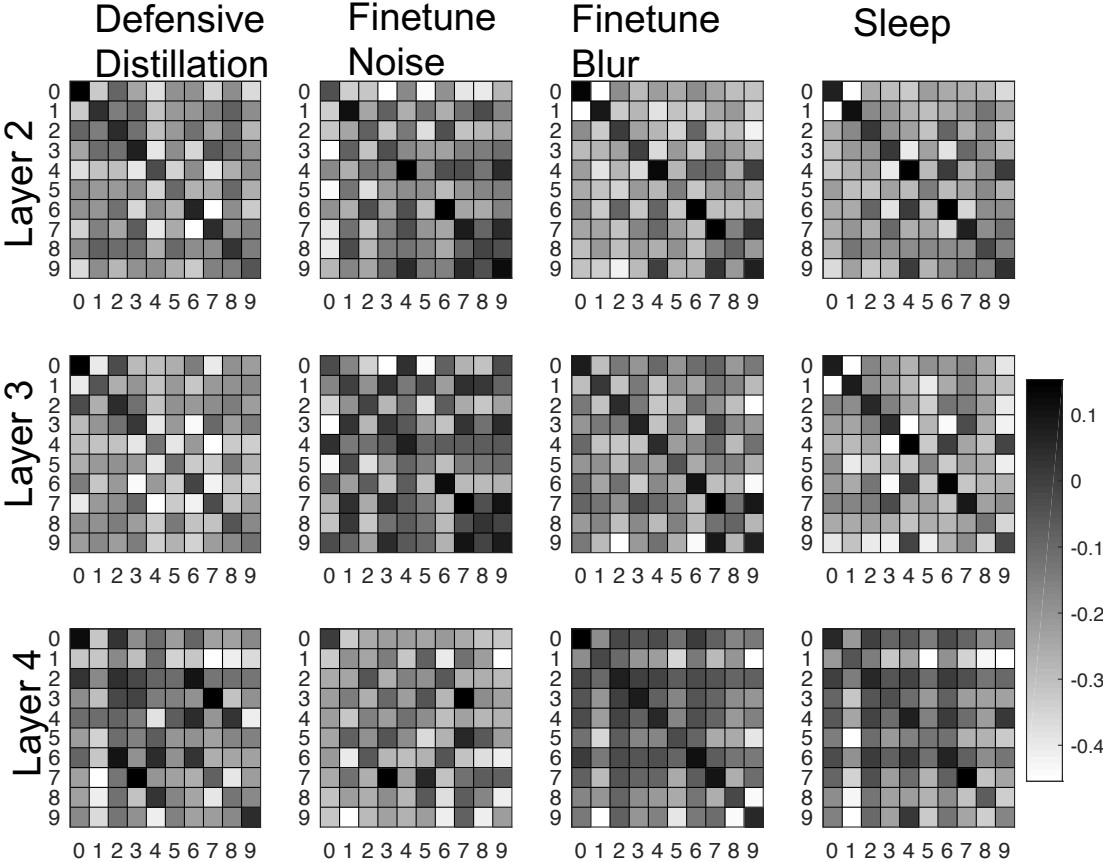

Figure 10: same as Figure 8 but activations are computed when presented with blurred images.

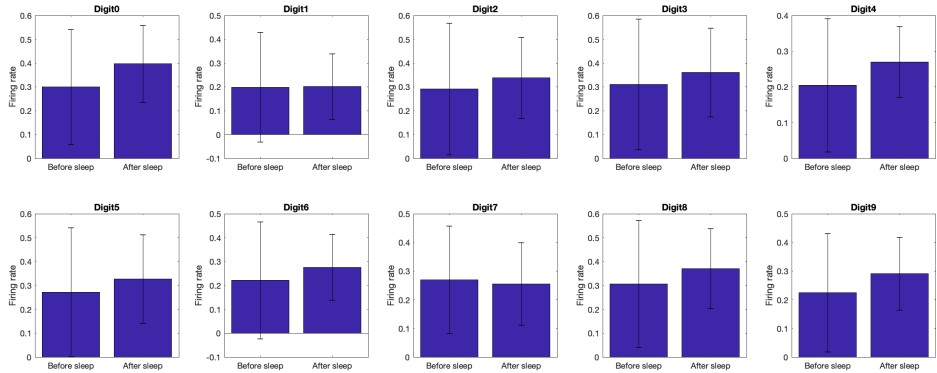

Figure 11: Normalized firing rates of neurons specific to individual digits when presented with noisy images is greater after applying sleep than before sleep.

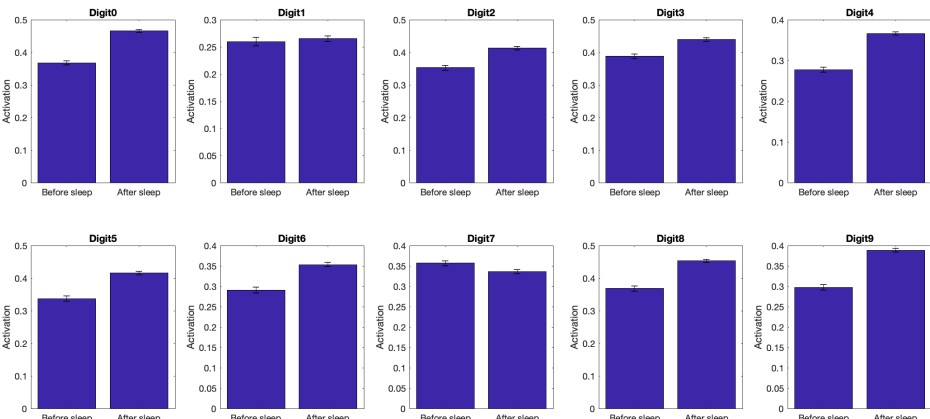

Figure 12: Normalized activations of neurons specific to individual digits when presented with noisy images is greater after applying sleep than before sleep.

