# OpenReview forum: "Biologically inspired sleep algorithm for increased generalization and adversarial robustness in deep neural networks"
_ICLR.cc/2020/Conference — Accept (Poster)_

### Official Review · AnonReviewer1 · 2019-10-20
**Official Blind Review #1**

**Rating:** 8

**Review:**

Disclosure on reviewer's experience: I am not an expert on adversarial attack methods or defenses, but I am well read in the general literature on robustness and uncertainty in deep neural networks.

The authors present a biologically inspired sleep algorithm for artificial neural networks (ANNs) that aims to improve their generalization and robustness in the face of noisy or malicious inputs. They hypothesize that "sleep" could aid in generalization by decorrelating noisy hidden states and reducing the overall impact of imperceptible perturbations of the input space. The proposed sleep algorithm broadly involves 1) converting the trained ANN to a "spike" neural network (SNN), 2) converting the input signal (pixels) to a Poisson distributed "spike train" where brighter pixels have higher firing rates than darker pixels, 3) propagating the neuronal spikes through the SNN, updating weights based on a simplified version of spike-timing-dependent plasticity (STDP), and 4) converting the network back to an ANN after the sleep phase has finished. They present a detailed comparative study spanning three datasets, four types of adversarial attacks and distortions, and two other baseline defense mechanisms, in which they demonstrate significant improvements (in some cases) of the sleep algorithm over the baselines.

The core concept behind the authors' work is novel and interesting, and the experimental design is thorough and well controlled. Although the results are (I would argue) somewhat mixed, they are nonetheless positive enough to encourage more work in applying "sleep" and other relevant ideas from neuroscience to the problem of robustness in deep neural networks. I have some questions and concerns which I will detail per-section below, but overall, I believe that this paper is a valuable contribution to the literature and should be accepted once the authors have made a few necessary revisions.

Section 1: Introduction

"We report positive results for four types of adversarial attacks tested on three different datasets (MNIST, CUB200, and a toy dataset) ..."

It's debatable whether or not the results from the CUB-200 dataset are positive. The sleep algorithm fails to outperform the baselines for each attack type (except for an almost negligible advantage in accuracy on JSMA) and barely even outperforms the control network in most cases (2/4 attacks it actually underperforms the control). I think the authors should consider rephrasing this statement to better reflect the actual results.

Section 2: Adversarial Attacks and Distortions

FGSM: The notation used here is somewhat inconsistent with the source paper. Goodfellow et al use epsilon to denote what I think the authors call eta, and call the second term, epsilon*sign(grad(J)), eta. Furthermore, the authors state that "this represents the direction to change each pixel in the original input in order to decrease the loss function." But this doesn't make sense. An adversary should want to *increase* the loss function enough to cause a misclassification. Goodfellow et al use this expression to formulate a L1-like regularization term and describe the training procedure "minimizing the worst case error when the data is perturbed by an adversary", which seems more sensible. This section should be rewritten to be more consistent with the source.

Section 3: Adversarial defenses

Regarding distillation: "We use T=50 to compare with the sleep algorithm"

The authors should elaborate a bit more on the reasoning for this choice. It seems very arbitrary.

Sectioin 4: Sleep algorithm

1. Algorithm 1: Why is line 9 inside of the for loop? It doesn't seem to be at all dependent on t. One would expect the input to only need to be converted once. Additionally, in lines 11-13, the l's in W(l,l-1) and similar should be unbolded. It's confusing that the format changes (unless I am missing something and it's actually a different variable).

2. Spike trains should be more rigorously defined, preferably with formalized notation. It's a bit unclear exactly what they are from the current text. Are they just parameters for a Poisson? Or outputs from a poison over T time steps? Or something else?

3. "weights are scaled by a parameter to induce high firing rates in later layers"
It would be good to include more details on this parameter, how the values are chosen, and the intuition behind this idea. I assume it's because of higher level feature representations in later layers of deep neural networks.

Section 5: Results

1. It's confusing that sometimes accuracy refers to classification accuracy and sometimes adversarial attack accuracy. I would recommend assigning a different name to the latter, or making sure that a qualifier precedes every reference to "accuracy" in this section.

2. In the second section of the results table (which is missing a label), why is the JSMA value for Defensive Distillation bolded? The distance measures for both the control network and for fine-tuning are higher. It seems like fine-tuning should be the one bolded.

3. Figure 1: caption is incorrect; it states "adversarial attack accuracy" and it should be "classification accuracy", otherwise the plots make no sense.

4. "we observe that in the Patches and CUB-200 dataset, sleep has beneficial results in moving the accuracy function above the other defense methods"
It should be noted that this is only true for eta < 0.1. After that, sleep and the control both converge to 50% accuracy. Also this sentence should be reworded to be less visual and more quantitative (e.g. sleep tends to have higher median accuracy scores than the other methods for eta < 0.1).

5. "We observe that performance continued to drop after a sufficiently large amount of noise was added"
More than that, the other methods converged to a small band of accuracy values; sleep continued to deteriorate. This is a significant difference. It would be a good idea to re-run this experiment with a binary classification problem (e.g. only two digits of MNIST) and see if this phenomenon still occurs. Then, the noisy sleep classifier predictions could simply be inverted to get improved accuracy scores.

6. In the analysis of JSMA, as noted before,, it's rather dubious to claim that sleep had any kind of significant effect on the attack success rate (or distance) for CUB-200. I would rewrite this section to better represent the results.

7. Figure 2 formatting: Legend is overflowing out of the first figure. Additionally, the legend colors should be made to match across all three figures, and the legend should either appear in all three (if necessary for some reason) or only in one.

8. Figure 2: The caption is incomplete and possibly incorrect. It's not clear why the first and last figures differ from each other, and the caption does not indicate this. The caption also only mentions two datasets, even though it says "for the following three datasets".

Appendix:

General formatting needs improvement. A lot of figures are off-centered, text misaligned, missing axis labels, etc.

**Experience Assessment:**

I have read many papers in this area.

**Review Assessment: Checking Correctness Of Derivations And Theory:**

N/A

**Review Assessment: Checking Correctness Of Experiments:**

I carefully checked the experiments.

**Review Assessment: Thoroughness In Paper Reading:**

I read the paper at least twice and used my best judgement in assessing the paper.

---

> ### Author Response · Authors · 2019-11-12
> **Response to Reviewer #1**
>
> We would like to thank the reviewer for the positive and helpful comments. Below we provide a point-by-point response and list the changes we made to the paper:
>
> Section 1 – Introduction
> We agree that the wording overstates our results. We have re-worded the sentence to read, “We analyze how robust our sleep algorithm is to four different types of adversarial attacks on three different datasets (MNIST, CUB200, and a toy dataset). For most conditions (MNIST, toy dataset), after a sleep phase was applied, the attacks consistently resulted in adversarial examples that were more distinct from the original input compared to the adversarial examples designed for the original (before sleep) network.”.
>
> Section 2: Adversarial Attacks and Distortions
> Thank you for pointing out the inconsistencies. We have changed all instances of eta to be epsilon, where appropriate. We have also changed the sentence to read "this represents the direction to change each pixel in the original input in order to increase the loss function." We mistakenly listed x’ = x – eta, in which case eta would be the direction that decreases the loss function, but both of these typos have been fixed now  to say x’ = x + eta.
>
> Section 3: Adversarial defenses
> The paper describing defensive distillation (Papernot et al. 2016) reports that temperature values between 20-100 result in adversarially robust networks. Thus, we chose T=50 as a starting point. It should be noted that this temperature value may not work best for our network architecture and we have not tried to optimize this value in our study. However, we believe it still serves as a good baseline to compare with our approach.
>
> We added the following sentence to this section:
> "Based on the previous work which found that temperature values between 20 and 100 effectively prevent adversarial attacks (\citet{papernot2016distillation}), we use a temperature value of $T = 50$ in our implementation of defensive distillation."
>
> Section 4: Sleep algorithm
> 1.	The spike train is dependent on t. At each time step, a new input spike train is computed (with statistics given by the poissonian distribution). We have updated the S variable to show its dependence on t. Please also see our response to the 2nd point below. In all instances, symbols of l inside W(l,l-1) have been unbolded.  We modified the algorithm part to incorporate the time dependence.
>
> 2.	Sorry for the confusion – in our work,  spike train is a binary vector representing whether or not a neuron spiked in a given time step, S(l,t). So, if the input layer has 784 neurons and, e.g., neurons 1 and 784 spike, then S(1,t) = [1, 0, …  0, 1] for a given timestep t (1 denotes the layer number). We have edited the text to reflect this. We also changed the term to “spiking activity” to make it clearer.
>
> 3.	The parameter we refer to here is the weights of synaptic connections referenced in Table 3. The values are chosen based on a parameter sweep aimed at optimizing generalization performance on a subset of the training set. This intuition is based on the known properties of sleep and effect of sleep on memory in biological systems: during sleep synaptic connections are elevated due to neuromodulatory changes that promotes reactivation of neurons representing past memories. We also increased the connection strength across layers to ensure spiking activity occurs at all layers of the network.
>
>
> Section 5: Results
> 1.	We’ve added a qualifier to all instances of accuracy to make it clear whether we refer to adversarial attack accuracy (how successful the attack is) or classification accuracy (how successful the network/defense is).
>
> 2.	The value for fine-tuning is now bolded.
>
> 3.	Yes, thank you. Indeed, we mean to say classification accuracy. This figure caption has been updated.
>
> 4.	Changed to “Looking at the classification accuracy of the network as a function of noise (epsilon, Figure 1), we observed that in the Patches and CUB-200 dataset, sleep tends to have higher classification  accuracy than the other methods for epsilon < 0.1. Beyond this point, sleep tends to have equal classification accuracies as compared to the other methods.”
>
> We continue the response in another comment.

---

> ### Author Response · Authors · 2019-11-12
> **Response continued**
>
> Section 5: results
> 5.	This is a good point which we believe is composed of 2 related questions:
> -Why don’t the accuracy values all converge to 0? Indeed, as more targeted noise is added (noise that increases the loss for a specific example), we would expect to observe more misclassifications until all examples are misclassified.
> -Why does sleep converge to a lower value than the other defenses (even if it converges more slowly)?
>
> In response to the first question, in the paper, we clip pixel intensities to be between 0 and 1 and use ReLU neurons. Both of these things cause convergence to be non-zero, since there is a hard limit on how much change can be observed (reaching the clipping bounds, or causing a neuron to go inactive by making its activation 0). Thus, the non-zero convergence is a result of not being able to change the response of the network any further. If we change the activation function of the output layer (to softmax for example), all defenses converge to 0 accuracy, with sleep converging the slowest.
>
> As for why, with ReLU activation functions, sleep converges to a lower value than the other defenses, we believe that this is due to an increase in activations following sleep. Since some activations are larger, there is more room to distort an input before it causes a neuron to go inactive (0 activity), thus resulting in a lower value of convergence. Again, with different activation functions, sleep actually converges the slowest since there is not artificial bound on how much one can distort an input before making certain neurons inactive, thus giving more room to increase the loss for a specific adversary.
>
> In the binary case, we found that all defenses converge to around the same value when we do not clip the input images.
>
> 6. We have modified this paragraph to accurately reflect the results as suggested:
> “Sleep successfully increases the network's robustness to the JSMA attacks for MNIST and Patches,  reducing the attack success rate in the case of MNIST and increasing the distance needed to create an adversary for Patches. On CUB-200, there is a marginal reduction in the adversarial attack accuracy compared to the control network. Defensive distillation and fine-tuning also reduce JSMA's effectiveness. However, for these two defenses, in the case of MNIST, the networks were capable of finding an adversary for a higher percentage of the testing set. Thus, the effect of changing a small number of important pixels is mitigated after running the sleep algorithm.”
>
> 7. Figure 2 formatting has been updated. Note that the legend for panel A does not apply to panel B since there are 5 networks in panel A but only 4 in panel B (since CUB-200 feature embeddings cannot be blurred). This has been mentioned in the caption as well.
>
> 8. The Figure 2 caption now reads:
> “Sleep increases robustness to general distortions. A) Generalization classification accuracy for 5 networks with noise and blur on the MNIST dataset. B) Generalization classification accuracy for 4 networks for the CUB-200 images with noise. Note that there are only 4 networks because there is no blur task here.”
>
> Appendix: We have updated the formatting to this section.
>
> Thank you for the these important comments which help to improve the clarity and presentation of our work.

---

### Official Review · AnonReviewer2 · 2019-11-02
**Official Blind Review #2**

**Rating:** 6

**Review:**

The paper proposes an ANN training method for improving adversarial robustness and generalization, inspired by biological sleep.

I'm leaning towards accepting as it seems to be an original concept and has fairly extensive empirical results that are somewhat promising.

The idea of a sleep phase as an alternative to explicit adversarial or generalization training is interesting. The results suggest that the approach works reasonably well in many cases.

Suggestions for improvement / clarification:
- The mapping from biological sleep to the actual algorithm + pseudocode used could benefit from more thorough explanation. It is not clear which choices are arbitrary vs well-principled.
- Was the optimal sleep duration determined empirically for each experiment?
- I agree with the authors' proposed future work of better understanding and standardizing this approach.
- Consider combining this approach with the existing adversarial or generalizing approaches (instead of as an alternative). Do they complement each other?

**Experience Assessment:**

I do not know much about this area.

**Review Assessment: Checking Correctness Of Derivations And Theory:**

I did not assess the derivations or theory.

**Review Assessment: Checking Correctness Of Experiments:**

I assessed the sensibility of the experiments.

**Review Assessment: Thoroughness In Paper Reading:**

I made a quick assessment of this paper.

---

> ### Author Response · Authors · 2019-11-12
> **Response to Reviewer #2**
>
> We would like to thank the reviewer for his/her efforts and time to review our paper. Below, we provide point-by point response to the reviewer’s comments.
>
> 1.	 Mapping from biological sleep to the actual algorithm
> We agree that this point could be further clarified. Deep sleep in brain networks is characterized by oscillations between periods of elevated activity (Up-states) and periods of silence (down-states). Both computational models and experimental studies have demonstrated reactivation, during Up states, of activity pattern observed during awake, which is often referred as “replay” (eg: Wilson et al., Wei et al.). Furthermore, evidence from biological work suggests that reactivation during sleep allows for generalization and schema formation (Lewis et al.). In this work, we developed an algorithm that emulates certain aspects of a biological sleep for artificial neural network to allow for spontaneous reactivation of memory patterns during sleep.
>
> To connect biological sleep properties to our algorithm, we would like to note the following:
> - The memory replay in biological sleep is stochastic. Since there is no input, the reactivated neuronal patterns may be different from those observed during awake state. Thus, the input we present to the network during sleep phase of our algorithm is also noisy.
> - Due to the effects of neuromodulators during sleep, excitatory connection strength is elevated, which is one of the critical factors contributing to synchronized activity during sleep. To account for this property, we scale the weights up during sleep in our algorithm during sleep (this is the Synaptic AMPA current in Table 3).
> - Asymmetric STDP: The nature of changes of synaptic plasticity mechanism during sleep is not well understood. Here we found that synaptic strengthening (the increase factor) being higher than synaptic depression (the decrease factor) allows for the best results. Hence, for most datasets the increase factor is greater than the decrease factor.
>
> We hope this provides a more thorough explanation of the connection between biological sleep and our algorithm. As for which decisions are arbitrary, we have yet to find a good way to compute thresholds for neurons during sleep without relying on a parameter sweep. Similarly, the sleep duration is determined solely by a parameter sweep and can be considered arbitrary.
>
> 2.	Parameter determination
>
> We used a genetic algorithm to identify the best parameters for sleep (parameters include sleep duration, thresholds for each layer, weight scaling factor, and plasticity rules - how much to increase or decrease the weights by). We chose parameters that optimized performance on the FGSM attack (on the training set). These parameters also give rise to good performance on other attacks, as well as the test set for FGSM, indicating the robustness of the approach.
>
> 3.	Future work
> As per understanding how the sleep algorithm gives rise to better generalization, from related work we can give the following rationale/hypothesis. In general, during sleep, there is a trend toward synaptic depression (in biology, this idea also has some support [1]). However, strong synaptic pathways through the network are usually unchanged or increased. In our algorithm, the pathways that are less important in representing a specific digit are reduced, in agreement with biology. This can lead to increased generalization by reducing the impact of the noisier pathways on classification. We are still working on verifying this hypothesis.
>
> 4.	Ensembling approach
> There are likely some other approaches for increasing generalization which complement our approach and could be used in conjunction. For example, from our Appendix, it looks like both defensive distillation and fine-tuning have similar effect as sleep – decorrelating the representation of different digits (Figure 8). However, the effect of sleep is more pronounced for noisier images (Figure 9-10).
>
> Fine-tuning may be complementary as well. However, without access to all types of noise that may be present during testing, fine-tuning will fail on types of noise it has not seen before. Thus, some combination of fine-tuning and sleep could be helpful in increasing accuracy (for example, fine-tuning on blurry images generally does better than sleep alone) while also increasing robustness to untrained types of noise.
>
> Thank you for all your efforts and your help in reviewing our paper.
>
> [1] Tononi, Giulio, and Chiara Cirelli. "Sleep function and synaptic homeostasis." Sleep medicine reviews 10.1 (2006): 49-62.

---

### Decision · Program_Chairs · 2019-12-19

**Decision:**

Accept (Poster)

**Comment:**

"Sleep" is introduced as a way of increasing robustness in neural network training. To sleep, the network is converted into a spiking network and goes through phases of more and less intense activation. The results are quite good when it comes to defending against adversarial examples. Reviewers agree that the method is novel and interesting. Authors responded to the reviewers' questions (one of the reviewers had a quite extensive set of questions) satisfactorily, and improved the paper significantly in the process. I think the paper should be accepted on the grounds of novelty and good results.